# Hugh MacDiarmid's 'On Raised Beach': 'Geopoetics' in a Time of Catastrophic Crisis' †

Richard H. Roberts

Faculty of Divinity, University of Edinburgh, Edinburgh EH8 9YL, UK; robertsrh28@aol.com
† This paper is revised from what was originally delivered as an on-line public lecture in November 2020.

**Abstract:** The poet Hugh MacDiarmid (1892–1978) was the major driving force behind the twentieth-century Scottish literary renaissance and was also a passionate Scottish nationalist. His poem 'On a Raised Beach' (1934) has been understood in theological and philosophical terms as a metaphysical exploration, albeit one grounded in an immediate experience of nature that took place on Shetland. In this paper, MacDiarmid's epic is placed in the context of the present environmental crisis and the ongoing consequences of the COVID-19 pandemic. 'On a Raised Beach' can now be re-located within the hermeneutical tradition of 'Geopoetics', a Scottish genre that is articulated and asserted by the poet Kenneth White (1938–). Whilst, however, White draws upon the highly contested and polyvalent concept of 'shamanism' in elaborating his standpoint, we shall argue that it is also appropriate to look for affinities between this dynamic poem and the ethos and mysticism of 'deep ecology', a perspective that invokes the equally contested mythology of 'Gaia'.

**Keywords:** Geopoetics; crisis; nature; pandemic; Hugh MacDiarmid; deep ecology; *On a Raised Beach*; shamanism; Gaia

**Inspiration**

> All is lithogenesis—or lochia, . . .
> Making mere faculae of the sun and moon,
> I study you glout and gloss, but have
> No cadrans to adjust you with, and turn again
> From optik to haptik and like a blind man run
> My fingers over you, arris by arris, burr by burr,
> Slickensides, truité, rugas, foveoles,
> Bringing my aesthesis in vain to bear,
> An angle-titch to all your corrugations and coigns,
> Hatched foraminous cavo-rilievo of the world,
> Deictic, fiducial stones.

From *On a Raised Beach*, Hugh MacDiarmid (1892–1978)

## Introduction: COVID-19 and the Climate Crisis: An Apocalyptic Disruption?

We are now living in a global context which is, in the classic Durkheimian sense, in a state of 'collective effervescence' (Liebst 2019). Movements are taking place in processes of spontaneous and imposed change that are *both* risk-fraught *and* essential to social transformation. The overall situation was significantly different when in late 2019 I was invited to deliver the annual Tony McManus Geopoetics Lecture.[1] My initial thoughts were based on the assumption that the world was still manifesting *homeostasis*, a self-regulating stability, albeit one threatened by the slow yet accelerating, multi-dimensional

environmental degradation associated above all with climate change. In 2019 there was, of course, a growing awareness that we were living in the era of the Anthropocene, in which the results of human activity, industrialisation, and the use of carbon energy sources are now being laid down in geological deposition.

In late 2019, global environmental activism was taking place inspired by such iconic figures as Greta Thunberg and David Attenborough. The highly focused gestural challenges of Extinction Rebellion (ER) were beginning to be applied at societal and economic pressure points. In investment and the global macro-economy, environment, sustainability, and governance (ESG) were gaining traction in global business. Simultaneously, an incipient aesthetic and affective yearning for close encounters with Nature in the face of its destruction was becoming prominent in the rediscovery of the genre of nature writing, and so on. Yet, despite these factors, the situation felt manageable, despite the rampant populism evident in, not least, the Trump presidency in the USA.

The onset of the COVID-19 pandemic has substantially destabilised the configuration that could be assumed in the autumn of 2019. Now, by contrast, we exist in what for us in the West, Europe, Anglo-America—and Scotland—is a time of acute crisis unparalleled in recent human history. The state of affairs that has come upon us in the interim is not of course a singularity, but a composite interpenetration of crises that are environmental, economic, and societal. The longstanding and difficult to define 'global problématique' (Ruggie 1980) has now been radicalised, intensified, accelerated, and thus rendered yet more complex.

This radicalisation and *intensification* of the human condition should be understood in terms of a total ecology, a 'human ecology' (Roberts 2005).[2] Our central concern in this paper is to explore the promise and limits of 'geopoetics' in the context of the crisis and its ongoing consequences. The COVID-19 global pandemic will oblige us to regard the years 2019–2020 as a major hiatus. In the light of this break, before and after the impact of COVID-19, how should we regard writing the world? How stands our life-world (*Lebenswelt*) in the Geo/Cosmos of Scotland? In what we shall shortly see what Kenneth White calls the movement from 'wording to worlding'? Our answer to this question takes us into many interconnected areas. The Scottish Geopoetics tradition affords a starting point for radical reflection and an innovative reading of a major poem. What at first might look in limited terms such as a revival of nature-writing in the face of ecological crisis is misleading. We shall argue that MacDiarmid's great if baffling poem may as *geopoesis* be construed as a prism, a microcosm through which the many dimensions of the macrocosm may be refracted.

## 1. Kenneth White and Geopoetic Impulse—'Earthing/Grounding a World'

As a distinctively and specifically Scottish genre, 'Geopoetics' has both a recent history and more remote antecedents. Whilst nature writing is undergoing a resurgence worldwide and while 'eco-theology' and 'deep ecology' spirituality thrive (Drengson and Inoue 1995), why should we adopt a hermeneutical context that is grounded specifically in 'geopoetics', a genre that is now enjoying an increasing salience both in Scotland and more widely across other continents? We shall see how this approach is distinctive: it is grounded in assertion and renewal, rather than in ceding to nostalgia and resignation.

The term 'geopoetics' was coined and has been developed as a life-long project by the poet and writer Kenneth White (1936–). White's wide-ranging, uncompromising, and at times inaccessible intellectual and poetic *oeuvre* has required a mediator. Much of White's extensive work was written in French and was published in France, and his recognition in Europe and internationally has not thus far been matched in Scotland. The late Tony McManus took up the role of interpreter in his posthumously published book *The Radical Field: Kenneth White and Geopoetics* (McManus 2007).

It would not be an exaggeration to regard Kenneth White as a prophet who remains largely un-honoured in his native land.[3] The reasons for this may not be immediately obvious to those lacking first-hand acquaintance with Scotland. There are well-established

paths to fame and prominence in Scotland, but the juxtaposition of White's birth in the Gorbals, an upbringing in Fairlie near Largs, a father who was a railway signalman, then a double first in French and German at the University of Glasgow, a state doctorate in Paris, and a life lived for extended periods in France and Europe appear to have placed him well outside the shared norms of the dominant Scottish poetic elite. There is an inherent resistance in some quarters to White's poetry, witnessed, for example, when he appeared at a Stanza poetry festival held in the University of St Andrews some two decades ago. The local distinguished poets in that ancient seat of academic endeavour were pointedly absent.

Kenneth White is an independent-minded outsider. Tony McManus shows us that White's work is the product of extended and unsparing creativity that is never out of border-crossings in nature, culture, nations, languages, and above all, in the intensification of place and moment. It is, however, only towards the end of McManus' book, *The Radical Field*, that we are finally given a definition, albeit one that is full and complex, of the term 'geopoetics'. This is a passage that is frequently deployed by White, and we cite it in full:

> Geopoetics is concerned with 'wording' (and 'wording' is contained in 'world-ing'). In my semantics, 'world' emerges from a contact between the human mind and the things, the lines, the rhythms of the earth, the person in relation to the planet. When this contact is sensitive, subtle, intelligent, you have 'a world' (a culture) in the strong, confirming and enlightening sense of the word. When that contact is insensitive, simplistic and stupid, you don't have a world at all, you have a non-world, a pseudo-culture, a dictatorial enclosure or a mass-mess. Geopoetics is concerned with developing sensitive and intelligent contact, and with working out original ways to express that contact. (McManus 2007, p. 183)[4]

At this point, it would be possible to tackle the fundamental questions that are evoked by this definition at some length.[5] Our goal is to show that Hugh MacDiarmid's epic poem 'On a Raised Beach' (1934) can be read and understood as an example of 'geopoetry', a *geopoesis,* and a 'world-making' of the kind that is later articulated by Kenneth White. Unlike White, Hugh MacDiarmid largely resided within his native land, and as a fervent nationalist, he was to become the controversial driving force behind the Scottish twentieth century literary renaissance. Yet, what both men share is an immediacy of direct (as opposed to recollected) experience, and it is this direct contact with the physical world that has an increasing salience in an era of complex, potentially catastrophic crises.

At this present juncture, the question as to what now constitutes what White calls 'sensitive, subtle and intelligent' contact with the world is now rather more complex than it might seem at first sight. We shall argue that whilst the salience of geopoetics may have increased, the paths of appropriation have become more fraught, arduous, and problematic.[6] This is apparent when we briefly review the present crisis.

## 2. COVID-19: An Apocalyptic Disruption?

On a pragmatic level, the continuing COVID-19 pandemic crisis is a disruption that is characterised by ongoing anti-globalisation, regressive nativism, recrudescent fascism, and riotous social media that is presently manifesting a range of ambiguous phenomena that are associated with conspiracy theories and panic thinking. At the core of this entanglement is the apparently uncontrolled proliferation of fear—and there is indeed much to fear. Yet, there is an ever-growing awareness that the great fear is not solely a spontaneous psychological, anthropological, and societal phenomenon, but is instead one that is fuelled and distributed through social media and the analysis and manipulation of big data. Conspiracy theories abound; people have found their friendships breaking as they locate themselves on different sides of the ideologically charged divide between saving the economy and saving lives.

Panic thinking, a short-circuiting of mental processes is now explained in a variety of ways, including through accounts of the misplaced operation of different parts of the brain when faced by imponderable and unassimilable experiences. Trauma is not marginal

but seemingly universal. The phenomenon of mass fear is not of course itself new, but its rapid, virtually instantaneous mediation arguably is. When events take place that exceed the previous bounds of shared sense, the mythic dimension comes to the fore. In traditional theological terminology, our present merits characterisation as a time of judgement and decision (κρισις), and even apocalypse (ἀποκάλυψις).

Resonant prior examples of the resurgence of apocalyptic ideas in the context of extreme crisis and change are evident in, for example, Edward Gibbon's famous account of the fall of the Roman Empire (1776–1789), accounts of the Black Death (1348–1353), Joachim of Fiore's idea of the Third Age (circa 1130–1201), the aftermath of the Napoleonic Wars (Napoleon represented by Nostradamus as the Antichrist), and, most relevant here, Oswald Spengler's cyclic account of the rise and fall of civilisations, *The Decline of the West* (*Der Untergang des Abendlandes* (Spengler 1923)[7], which was published during the period of disruption after the First World War in Germany (Roberts 1992).

The ambiguous return of notions of eschatology, apocalyptic, and associated panic thought invite clarification in terms of secularisation and re-mystification theory, and indeed psychopathology; once more, however, though relevant to our topic, these factors are not our primary concern. Here, we are not engaged with the sociological or anthropological explanation of the generation of myth and apocalypse as such[8] but with the more pragmatic question as to what in our extraordinary present time do Geopoetics and MacDiarmid's epic *On a Raised Beach* have to offer us as a basis for reflection and guidance for the conduct of life in a small country and in a world beset by multiple challenges? The latter include the underlying dialectic of the real and the virtual, which has ontological implications for a new, context-relevant understanding of the 'Caledonian Antisyzygy' (Smith 1919)[9] and thus the renowned 'divided mind' of Scotland.

### 3. On a Raised Beach in Context

My first encounter with the Scottish geopoetic impulse took place half a century ago in Cambridge under the tutelage of Professor Donald M. MacKinnon (1913–1994), then Norris-Hulse Professor of Divinity and at the height of his brilliant yet sometimes eccentric powers. In his demanding and fascinating lectures on theology and logical analysis, Mackinnon introduced his hearers to MacDiarmid's *On a Raised Beach*, a poem that was then relatively little-known in England. MacKinnon, a highly influential teacher,[10] regarded this text as one of the greatest metaphysical poems of the twentieth century. This extraordinary poem has a stunning intensity and displays a virtuosic juxtaposition of recondite vocabulary that is allied with the poet's paradoxical ambition to renew language.[11]

Each return to *On a Raised Beach* has moved me in different ways. For a start, there are intertextual resonances between Christ's forty days in the wilderness and *On a Raised Beach*. Born in Langholm in the Scottish Borders, Christopher Murray Grieve (later Hugh MacDiarmid) was raised in a culture that had long been permeated with both the Geneva Bible and the Authorised Version, and he knew the Bible well. This is a significant factor in any informed approach to the poem, yet it pertains to but one layer in a polyvalent text. There are many biblical allusions and a theological (or anti-theological) critique running through much of MacDiarmid's oeuvre (Whitworth 2007). On returning this time to *On a Raised Beach* in the context of the COVID-19 pandemic, the following question arose for me: What might be an appropriate hermeneutic to deploy when seeking to correlate MacDiarmid's distant poem with our fraught context and our correlative need to inform and nourish the human resolve to survive?

Long before I became aware of MacDiarmid, I had grown up with a passion for geology, both as the study of the rocks and as a key to securing a grounded identity. In the context of a childhood passed in the north of England in the decade immediately following the Second World War, it was, however, an English geopoetic text that attracted my attention. This book was *A Land*, which was written by the pioneer woman geologist and archaeologist, Jacquetta Hawkes (1910–1996) and was first published in 1951 (Hawkes 1951). In a recent re-review, Robert MacFarlane described *A Land* as in parts:

a short history of Planet England; a geological prose-poem; a Cretaceous cosmi-
comedy; a patriotic hymn of love to Terra Britannica; a neo-Romantic vision of the
countryside as a vast and inadvertent work of land-art; a speculative account of
human identity as chthonic in origin and collective in nature; a homily aimed at
rousing us from spiritual torpor; a lusty pagan lullaby of longing; and a jeremiad
against centralisation, industrialisation and "our" severance from the "land".
(Macfarlane 2012)

Macfarlane highlights Hawkes' 'ecstatic holism', and it is this primal sensibility which
would appear at first glance to connect a quintessentially English writer with the militant
Border Scot Hugh MacDiarmid and his geologically determined irruption.[12] Although it is
plausible to regard Jacquetta Hawkes (or even Kathleen Raine 1908–2003) as holistic 'Neo-
Romantics', MacDiarmid does not fall into this category. As a modernist inspired by James
Joyce's masterpiece *Ulysses* (1922) and in competition with T. S. Eliot and *The Wasteland*
(1922),[13] MacDiarmid's ambition, his grand project, certainly included reconnection with
the land, but what he ultimately envisaged was the revolutionary and highly controversial
renewal of an entire oppressed and blighted nation.

Regarding their affinities, Hawkes and MacDiarmid were both inspired by the act of
lying on the land. Correspondingly, both writers can be understood in terms of a somatic
epistemology, a body-centred perspective which is now widely theorised in ecophilosophy
(Fox 1990; Naess 1991) and ecopsychology (Roszak et al. 1995). In terms of the ethos of each
writer, Jacquetta Hawkes wrote as an upper-class Englishwoman living in the dying glow
of the British Empire. Hugh MacDiarmid the Borderer represented a nation that had been
crushed and forcibly domesticated by its neighbour after the Battle of Culloden (1746) but
which had then colluded with the Empire and its rich rewards (for some) of wealth, power,
and influence. This inner conflict between defeat and profitable incorporation is but one
element in the contradictory matrix of the Caledonian Antisyzygy.

With respect to the geopoetic turn, there are further tensions between these writers
and their traditions that resonate with the contrasting geologies of the British Isles. England
is sedimentary, whereas Scotland is as it were volcanic and orogenic in its identity.[14] We
can represent this difference in outlook and goals as a contrast between a layered English
'ecstatic holism', which displays affinities with the emergent subdisciplines that embody
ecological insights as opposed to a Caledonian intensity directed towards national renewal.

There is a further and important contrast to be drawn between Hawkes' English
*expressivist* ethos and an aesthetic traceable back, amongst others, to William Wordsworth
on the one hand, and, on the other, MacDiarmid's Scottish *expressionist,* confrontational,
even psychically violent breaking-free from any layered nostalgia. Seen in a broader context,
MacDiarmid's life-task was to assert a future-orientated prolepsis, a secular coming of the
Kingdom, the achievement of the full emancipation of Scotland from its subaltern status.[15]
The struggle continues in a divided nation.

## 4. Hugh MacDiarmid: Geopoesis and *On a Raised Beach*

In the explosive opening of *On a Raised Beach*, MacDiarmid[16] is at rhetorical odds with
any English rhapsodic nostalgia for lost connection with the land. What follows springs
out of touch, the pressure of opposed fingers holding a mere pebble, a symbol that is to
open access to eternity.

> All is lithogenesis—or lochia,
> Carpolite fruit of the forbidden tree,
> Stones blacker than any in the Caaba,
> Cream-coloured caen-stone, chatoyant pieces,
> Celadon and corbeau, bistre and beige,
> Glaucous, hoar, enfouldered, cyathiform,
> Making mere faculae of the sun and moon,

I study you glout and gloss, but have

No cadrans to adjust you with, and turn again

From optik to haptik and like a blind man run

My fingers over you, arris by arris, burr by burr,

Slickensides, truité, rugas, foveoles,

Bringing my aesthesis in vain to bear,

An angle-titch to all your corrugations and coigns,

Hatched foraminous cavo-rilievo of the world,

Deictic, fiducial stones. Chiliad by chiliad

What bricole piled you here, stupendous cairn?

What artist poses the Earth écorché thus,

Pillar of creation engouled in me?

What eburnation augments you with men's bones,

Every energumen an Endymion yet?

All the other stones are in this haecceity it seems,

But where is the Christophanic rock that moved?

What Cabirian song from this catasta comes?

From *A Raised Beach* (1930), Hugh MacDiarmid

What are we to make of this extraordinary passage? It is obviously virtuosic in that most of MacDiarmid's readers will have to reach for a large dictionary in order to understand the recondite vocabulary. Investigation of this passage may well expand the reader's *Wortschatz*, but contemporary interpretation does not depend upon knowing the meaning of all of these geological allusions and images. Having, however, tracked down the latter, it becomes apparent that whilst some expressions can be regarded as a demonstration of etymological ingenuity, an improvised verbal cadenza, other terms are fully germane to the greater project.

This most uncompromising opening is best understood for present purposes as a rhetorical tour-de-force that generates an acute *Verfremdungseffekt*[17] in the reader in a Brechtian manner. This passage may well provoke a convulsive response, a turning away, on the part of those who might shrink from the demands of the initial etymological investigation. For those who persevere, this passage could be likened to a verbal electric shock treatment that has been applied with benign purpose. The sheer power of the opening reminds the present writer of the persuasive effect of, for example, Germanophone writers such as Ernst Bloch (Roberts 1990) and Karl Barth (Roberts 1992), who built worlds through words that dislocate and who strove to create the equivalent of a numinous encounter.

Set in the context of the whole poem, commentators have been puzzled by this opening. The opening is a fiercely universal constative:

'All is lithogenesis—or lochia'

Literal paraphrase leaches out the compressed intensity of this line; but when deciphered and expanded, the shock of what lies behind these two obscure words of Greek origin, 'lithogenesis' and 'lochia', opens up a possible line of interpretation:

'Everything is born of or through stone—or the vaginal discharge of cellular debris, mucus, and blood following childbirth'

This uncompromising and shocking use of the birth image may be understood today as a strange prefiguring of Monika Sjöö and Barbara Mohr's chthonic feminism in *The Great Cosmic Mother* (Sjöö and Mohr [1987] 1991), and it resonates with (for example) the celebration of menstrual cycles in such contexts as the contemporary Red Tent movement.

The assertion of the axiomatic universal of a *geopoesis*, literally an 'earth-making', through birth and the female 'imaginary' (*imaginaire*) (Sartre 2010)[18] is no casual metaphor, but it may be construed as both a prescient ontological reversal and the prelude to identification with the material world of the most radical kind.[19]

In short, MacDiarmid, thus read and decoded, asserts an ontology that implicitly subverts what was the much-commented patriarchal normality of Scottish poetics.[20] This past norm is classically represented in Alexander Moffat's (Moffat 1980) famous painting, *Poet's Pub*, now in the National Gallery of Scotland, a canvas in which male poets encircle and venerate the master, whereas women are shadowy, marginal figures lurking on the periphery, apparently there but to service male requirements.[21]

Limits on space preclude a full analysis of this remarkable opening passage, but there is one phrase out of the eighteen-line lava-flow of words upon which we should focus. The opening effulgence culminates in the question:

What bricole piled you here, stupendous cairn?

And it evokes the response:

I study you glout and gloss, but have

No cadrans to adjust you with, and turn again

From optik to haptik and like a blind man run

My fingers over you,

This is an expression of total bafflement, an *aporia,* the sound and shock of the spade turning as it strikes bedrock. The move from 'optik to haptik' is critical: it amounts to a complete sensory shift from sight to touch in a benign somatisation. The touch of fingers upon stone is as pregnant in its significance as Marcel Proust's transition from the mundane into sensual memory at the outset of the monumental *A la Recherche du Temps Perdu*[22]. The smell and sensual qualities of the *madeleine* trigger Proust's life-memory and the great unfolding expressed in exquisite French. Interestingly, Proust explicitly alludes to a shamanic resonance elsewhere:

I feel that there is much to be said for the Celtic belief that the souls of those whom we have lost are held captive in some inferior being, in an animal, in a plant, in some inanimate object, and so effectively lost to us until the day (which to many never comes) when we happen to pass by the tree or to obtain possession of the object which forms their prison. Then they start and tremble, they call us by our name, and as soon as we have recognised their voice the spell is broken. We have delivered them: they have overcome death and return to share our life. (Proust 1913)[23]

The touch of the pebble between MacDiarmid's fingers does not trigger life-memory as such, but a sudden transition into both historical—and deep, primordial time. MacDiarmid's verbal eruption demands decoding, but despite the ambiguous outcome of etymological investigation, we suggest that the whole point is that we should not understand on first or even later readings: we are to be smitten and enter a process to which we submit ourselves when we engage with the whole poem. The analogue that comes to mind with regard to this passage is the way that a Zen master strikes the shoulders of the initiate meditator in a *keisaku* or utters a *katsu* (sudden shout) at the request of the student.

As we read later, the veil has been torn, the portal opens, and the poet enters a place outside of time.

　　Nothing has stirred

Since I lay down this morning an eternity ago

But one bird. The widest open door is the least liable to intrusion,

Ubiquitous as the sunlight, unfrequented as the sun.

The inward gates of a bird are always open.

It does not know how to shut them.

That is the secret of its song,

But whether any man's are ajar is doubtful.

I look at these stones and know little about them,

But I know their gates are open too,

Always open, far longer open, than any bird's can be,

That every one of them has had its gates wide open far longer

Than all birds put together, let alone humanity,

Though through them no man can see,

No man nor anything more recently born than themselves

And that is everything else on the Earth.

I too lying here have dismissed all else.

Bread from stones is my sole and desperate dearth,

From stones, which are to the Earth as to the sunlight

Is the naked sun which is for no man's sight.

I would scorn to cry to any easier audience . . .

Obviously, it would make sense at this juncture to move further into dialogue, or more accurately, a wrestling with the entire masterpiece of *On a Raised Beach*. Once more, this is not the time or place for such endeavour. Thus what, given our present purpose, are we to make of this extraordinary opening? We suggest that MacDiarmid is seeking to realise in text the long and transformative moment he spent in an altered state of consciousness (ASC) on the raised beach in Shetland. As we have seen, MacDiarmid has implicitly rejected any soft 'sublime' in visual imagery in the manner of a Wordsworth and even in the mature and recapitulatory Wordsworth of *The Prelude*.

The initial constative, the single, short, pungent sentence, 'All is lithogenesis or lochia' is then followed by what in classical rhetorical terminology is intensification, an *amplificatio* (*Steigerung*) that is enacted through a series of incantatory verbal blows using words of recondite origin. This demonstration of lexicographical range is not the product of a misplaced elite egoism of the kind that might well have a contemporary British university teacher reprimanded in a staff–student committee, but the annunciation of a path into an 'instasis' as opposed to 'ecstasis' (ἔκστασις): this is the inward irruptive thrust.

If we now move from the oriental parallel of the Zen shock and place the passage in a setting that is perhaps more familiar so that the verbal blow of deconstructive encounter may then also be construed as an analogue of the proleptic 'death' of shamanic initiation[24], which presages a journey beyond mundane confines. In *The Radical Field*, McManus draws our attention to Kenneth White's concern with the 'larger role' of the shaman[25] and to a 'journey' that makes profound connections of wide societal relevance:

> '(T)he shaman maintains the contact between the socio-human context and the world, the universe at large'. (McManus 2007, p. 73)

Both Hugh MacDiarmid the poet and Donald MacKinnon the philosopher–theologian strove to bring about such benign death-dealing and life-giving connections: they pruned in order to establish grounding and to stimulate deeper growth.[26] The words MacDiarmid deploys may trigger an alienation of consciousness, a *Verfremdungseffekt* in the reader. This is no comfortable familiarity but is instead a making-strange that renders the unfamiliar known. As such, it potentially allows access to a psychosomatic process: what has been triggered by touch may now open access to the *limen* (threshold). Before exploring this further, we briefly visit the historical context of this Caledonian drive towards immediate connection.

### 5. Prevenient Dialectics: Caledonian *Haecceitas* from Duns Scotus to the Travails of Modernity

What we have called MacDiarmid's *instasis* (as opposed to *ecstasis*) has affinities with Gerard Manley Hopkins notions of 'inscape' and 'instress'. Significantly, both poets resonate with the residue of the Scottish philosopher theologian Duns Scotus (circa 1265–1308) and his concept of dynamic *haecceitas.* (Brodie 1995). MacDiarmid was of course no priest, whereas Gerard Manley Hopkins was a Jesuit, yet both share traditions that evolved from Scotus with very different organisational outcomes. 'Haecceity' is a literal translation of the equivalent term in Aristotle's Greek *to ti esti* (τὸ τί ἐστι) or 'the what (it) is' from the Latin *haecceitas*, which translates as 'thisness'. Alexander Brodie's (1995) work is definitive in terms of Scotus in Scotland. In his indispensable *Gifford Lectures, The Shadow of Scotus; Philosophy and Faith in Pre-Reformation Scotland*, Brodie emphasises this focus on particularity when he cites Gerard Manley Hopkins' sonnet 'Duns Scotus Oxford' (Brodie 1995, p. 7). Whereas 'haecceity', the English derivative of *haecceitas,* can be defined simply as 'the property that uniquely identifies an object', we touch upon complex debates in scholastic philosophy and theology. The debates not only pass through the processes of secularisation, but through earlier transitions, notably the early Christian struggle with paganism. Underlying Christian traditions, both East and West, there lies the suppression of paganism and the correlative and longstanding problem of the construal of 'nature' (*thusis/natura*).[27]

In Scotland, the theological and ecclesial aporias are particularly acute, and this is despite the presence and power of outstanding landscapes and complex geology.[28] The distinguished diplomat and author John Buchan (1870–1940) refracted the past and anticipated the future, as did MacDiarmid. In his controversial historical novel *Witch Wood,* which was published in 1927 (Buchan 1927), Buchan depicted a post-Reformation Scotland that was fraught with intense religious conflict that was underlaid by the pervasive repression of what Carl Gustav Jung articulated as *Anima*, the female archetype. Similar to James Hogg in his haunting *The Private Memoirs and Confessions of a Justified Sinner* (1824), in *Witch Wood,* Buchan explores aspects of 'the shadow' in ancestral Scottish religious history.

A humane, lower key, and almost despairing counterpoise to both MacDiarmid and Buchan is Edwin Muir's (1887–1959) desolate poem *Scotland 1941*. Muir directly attributed this Scottish alienation from nature to the Reformation:

> We were a tribe, a family, a people.
>
> Wallace and Bruce guard now a painted field,
>
> And all may read the folio of our fable,
>
> Peruse the sword, the sceptre and the shield.
>
> A simple sky roofed in that rustic day,
>
> The busy corn-fields and the haunted holms,
>
> The green road winding up the ferny brae.
>
> But Knox and Melville clapped their preaching palms
>
> And bundled all the harvesters away,
>
> Hoodicrow Peden in the blighted corn
>
> Hacked with his rusty beak the starving haulms.
>
> Out of that desolation we were born.

If we draw together Hugh MacDiarmid in *On a Raised Beach*, John Buchan's *Witch Wood,* and Edwin Muir's quiet despair in *Scotland 1941*, it is apparent that the renewal of Scotland is an alluring, yet fraught and problematic project.[29] As an oppressed people subsisting in a historic and contested *Knechtschaft* vis à vis the *Herrschaft* of her southern neighbour, Scotland is perhaps better equipped to deal with her illusions than England (Roberts 1988).[30] Yet, the geopoetic adoption of the shamanic eye, and indeed a fully reawakened total sensorium,[31] afford insight into both pathology and promise, out of which

it might be possible to break ([Boyarin 2004](#)). Such a full endeavour would transport the reader beyond the shamanic encounter through the portal of somatic intensification into the debates and practices of non-dualism. It is the boundaries that all three writers transgress, the borderlands they inhabit and explore, and the latent potency of the Caledonian collective unconscious that provides a bridgehead.

If we represent Hugh MacDiarmid, John Buchan, and Edwin Muir in different yet related ways that serve as the precursors of the formulation of the specific geopoetic impulse and its drive towards a material authenticity, then we can see that at the very least, contemporary developments have important antecedents. Regarding the present and the future, this combined grounding in the past now permits us to outline the resonance between geopoetics and the COVID-19 pandemic and the climate crisis, a situation in which we are all in various ways both victims—and implicated—in a more exploratory way.[32]

We now briefly outline three recent texts that we believe resonate with MacDiarmid's liminal experience and his unsparing evocation of human origination. This implies that MacDiarmid was, as it were, touched by Gaia, a hypothesis and contention that is worthy of further exploration.[33] At this juncture, we propose the exploration of the elective affinities between a poetic tradition that asserts its shamanic character and varieties of response to ecological crisis in which the image of 'Gaia' has emerged as what amounts to another 'essentially contested concept' ([Gallie 1964](#)). Strangely, perhaps, mythic thought has re-asserted itself across the board in all four examples.

*a.*　　*John Seed and Joanna Macey: The Council of All Beings (1988)*

Our first example is the collection *Thinking Like a Mountain—Towards a Council of All Beings* ([Seed et al. 1988](#)), an influential deep ecology text that was edited by the Australian rainforest activist John Seed and the Buddhist activist Joanna Macey. This book advocates experiential regression to ultimate origins in the formation of the Earth, identification through grief and mourning with threatened or extinct species, and the possibility of psycho-spiritual rebirth through an encounter with Gaia. As part of his fieldwork, the present writer undertook participant observation in a three-day deep ecology workshop that was facilitated by John Seed in 1999 that was based on practices that had been drawn from this text. Not least, this experience made him aware that deep ecological neo-shamanic and ritual processes of the kind that are advocated by Seed and Macey may expose repressed current and ancestral trauma. Regression and psychic rebirth through Gaia are not for the faint-hearted.[34]

MacDiarmid's epic poem can be regarded as an experiential parallel to neo-pagan and deep ecological immersion, an opening in the contemporary religio-spiritual field that is represented in its most explicit form as the solitary 'vision quest'.[35] After due preparation, the vision quest is time spent alone in nature, during which the normal constraints, supports, and distractions of mundane responsibility are absent or at least weakened. The progressive stripping away of activities and normal defences combined with regression may well have powerful consequences. As a planned activity, the vision quest or ecopsycological retreat involves preparation, separation, time at the *limen* or threshold, and then return and reaggregation. In contrast, it would seem that the societal enclosure and isolation of entire populations during the 2020–2021 COVID-19 pandemic lockdown and the imposition of ongoing uninterpreted constraints have enjoyed no such preparation or framework. This lack of context and framework might well go some way to account for the exposure of trauma and mental health issues during the pandemic.[36]

Within the frame of reference of shamanic or charismatic psycho-spiritual processes, the societal effervescence that is associated with such phenomena may be the price paid for transformation, but this is controversial. In his classic introductory study of charisma, the anthropologist Charles Lindholm excludes this option. For Charles Lindholm and others ([Lindholm 1990](#); [Meštrović 1997](#); [Bowker 1987](#)), modernity fails to provide the supportive community in which the shaman/charismatic may safely operate. A hard road lies ahead for those who recognise the reality of the collective unconscious, the presence

of repressed shadow, and the necessity of integration on the path to both individual and societal maturation.[37]

b.　*Bruno Latour's Gifford Lectures (Latour 2017)*

A second example of the appearance of the image of Gaia is in the sociologist and anthropologist Bruno Latour's Gifford Lectures, which were published under the title *Facing Gaia: Eight Lectures on the New Climatic Regime* (Latour 2017). Unlike Seed and Macey, and of course in a different genre and on a magisterial intellectual scale, Latour hesitates to invoke Gaia as a contemporary spiritual resource. In his Giffords, Latour provides a fascinating and complex account of the climate crisis that is conceived under the rubric of the 'great acceleration' and develops a project that dares to confront the 'curse of Gaia'.[38]

Whilst the image and origins of Gaia in the fragments of Hesiod are touched upon by Latour, he hesitates to proceed on a mythopoeic path because of the violence and chaos that are associated with this mythology in archaic Greek thought. The title of the third lecture, 'Gaia a finally secular figure for nature', is indicative. Latour's hesitance and his avoidance of any quasi-theology of Gaia contrasts with the deep ecological acceptance of Gaia as a symbolic and experiential resource. Latour's move also heads off James Lovelock's recent revisiting of the Gaia hypothesis, in which, as we shall shortly see, he welcomes a catastrophic 'ecological eschaton'.[39] Under the rubric of the Gifford Lectures, Latour has, as have many before him, extended and interpreted the meanings of 'natural theology'.[40]

c.　*The Planet in a Pebble: Scientific Geology*

After the poets and activists and then the anthropologist, the geoscientist Jan Zalasiewicz in *The Planet in a Pebble: A journey into Earth's deep history* (Zalasiewicz 2012) provides a third example of earth-encounter and the potential of geopoetics as tool in recovery. Whilst Zalasiewicz propounds straight geological science, the implications of the advent of the Anthropocene and the deployment of the concept of deep time take his argument beyond the pre-existent relative homeostasis to which we referred to at the outset of this paper. The title of this book, *The Planet in a Pebble*, ventures beyond the literal and inadvertently resonates with MacDiarmid's moment of transition and transformation.

There is thus a limited parallel and resonance between the geoscientist's admission of a mythic dimension as a means of conveying the extremity of this era of crisis and MacDiarmid's recourse to geological particularity and to the relevant science from which he not only drew images and built into his text a process that we might describe as an early neo-shamanism. Whilst MacDiarmid indubitably displays linguistic virtuosity, his use of geological terminology is not arbitrary; he knows and emphasises the sheer diversity and beauty of the rocks and geological structures of Scotland. All of these examples point to affinities; there are, however, some major issues that point to difference and threats.

d.　*Gaia and the human parasite*

In the early stages of the pandemic in the United Kingdom, the problematic doctrine of 'herd immunity' (Jones and Helmrich 2020; Grover 2021) was invoked and contemplated, albeit in a covert way. The recent tendency to allow for an uneasy alliance between the largely implicit assumption that the 'hidden hand' will somehow provide and that a Malthusian attitude to human survival is ethically acceptable gained some traction in the public arena. It is, however, James Lovelock, our fourth exponent of Gaia, who strips away any residual sentimentality about human survival in a stark mythic catastrophism. Lovelock's (2019) seminal account and later development of the Gaia Hypothesis represents now humanity as the 'human plague' afflicting the Earth. This is a highly controversial vision of humanity as an illness:

> Just as the human body uses a fever to fight off an infection, Gaia is raising Her temperature to expel a harmful parasite—humans. Unless humans renounce their destructive ways and rejoin the diverse community of living beings in Gaia's loving embrace then Gaia will be forced to act in order to secure Her supreme reign.[41]

In the setting of such a vision, the COVID-19 pandemic manifests as the act of Gaia as cosmic self-protection, and this suggestion has evinced a ferocious response in the conspiracy zone on the web. Nevertheless, there is an extremely serious point informing Lovelock's notorious declaration. The key words are 'renounce' and 'rejoin': how might such renunciation and rejoining be furthered? Such questions have invited worthy, yet now sometimes seemingly lightweight, answers around conservation, re-wilding, greening business, and so on. These piecemeal responses that have survived in the era of assumed homeostasis; now, however, it would now appear that the crisis requires far more than what the highly vocal activist Greta Thunberg has decried as the 'blah, blah, blah' of climate negotiations. In short, the human parasite should not kill its host; yet, how to achieve this implies full-scale global action of the kind demanded but not fully achieved at COP26.

Before drawing these reflections towards a conclusion, there is yet a further dimension to be touched upon that affects both contemporary readings of MacDiarmid's epic and, more widely, changes in the categorial assumptions that inform philosophical and theological discourse through millennia and across cultures. Embodiment, somatic knowledge, and the journey to the limen are affected by changes in human experience and in our understanding of the categories of space and time.

## 6. New Dialectic I: Virtuality, Body—And 'POST-Humanity'

There is a major tension between this reading of *On a Raised Beach* and Kenneth White's assertions about 'shamanism' on the one hand (White 2003)[42] and the construal of the highly contested concept of 'modernity' on the other. This can be characterised on one level as a crude contrast between aspirant individuals of groups who seek to penetrate the veil and access the *limen* and those such as Richard Dawkins, who propound and popularise a rational and informed scientific worldview in, for example, *The God Delusion* (Dawkins 2006). Such a juxtaposition is of course misleading in that, for example, cognitive neuroscience exposes in extensive literature the complexity rather than the illusory nature of spiritual experience (Hick 2010; Sayadmansour 2014).

Beyond the level of polemical strife, both the assertion of a neo-shamanic approach and scientific (or scientistic) positivism need to take account of two emergent new dialectics in relation to space and time. These factors pertain to the changing accounts of the manifold of the categories that have, since the time of Aristotle and much later Hume, Kant, Hegel, and not least Marx, served to make philosophical sense of the structure of human self-understanding and world-construction. Such concerns were central to the thinkers of the Enlightenment, be they in Scotland, Europe, or in North America.

Whilst the 'otherworld' of the shaman practitioner and notions of transcendence in mainline non-atheistic religions or spiritualities display affinities in terms of their admissibility of concepts of transcendence, both would appear to be threatened by the all-embracing ambition of technology (Chopra 2021).[43] Human dependence upon virtual reality (VR) and mixed reality (MR) has expanded under COVID-19 conditions in an extraordinary way. Simultaneously, and paradoxically, the drive for embodiment has proliferated and intensified in the interlinked global spirituality and wellbeing industries (Cook-Cottone 2015). These contrasting virtual and somatic drives compete with each other in the complex and ambiguous struggle for embodiment. In the growing field of the philosophy of virtual reality (VR) and cyborg culture (CC), it has become apparent that ancient conceptualisations of forms of transcendence pertaining to the individual, and reflected upon since the pre-Socratics in the West and equally early in Vedic thought, have in effect been absorbed and operationalised by technology (Heim 1994; Metzinger 2018).

Whilst some categories in traditional philosophy and theology are now rendered problematic by the availability of highly effective surrogates in VR and CC, the latter have plundered the former for the some of the conceptuality used to represent the enlargement of human capacities—or their displacement by artificial intelligence (AI). The tension between VR/MR and the drive towards embodiment and its consequences have been immeasurably radicalised by COVID-19. As virtual and cyborg cultures expand and

proliferate, the masking, lockdown, and isolation measures that have been imposed to combat the COVID-19 pandemic further challenge and limit the very possibility of enjoying a grounded and embodied existence. Such existence is arguably fundamental to the creation of the *communitas* explored by Arnold Van Gennep and Victor and Edith Turner and others in studies of the ritual process. In sum, what human beings really need, the emergence of a rich convivial community out of *communitas*, becomes even more difficult to achieve.

Under the conditions of the COVID-19 pandemic, the absence of viable categories of interpretation and the fragility of reliance upon information technology magnify the likelihood of panic thinking and the urge to escape into fear-stoked fantasies. Fact and fake are easily confused and are therefore highly manipulable. Under these conditions, the relentless facticity, the sheer *haecceitas* of the stone held and felt in the hand, grounds and anchors us both in the present. It allows us to connect with the poet who long ago lay on the raised beach: furthermore, our capacity to enjoy touch draws MacDiarmid's prescience into the demands of our collective present. At the very least, a stone held between fingers is an exertion of sober sense.[44]

### 7. New Dialectic II: Time, Acceleration—And Economy

Adam Smith's epochal theorisation of the intellectual structure of the emergent industrial system in political economy and his articulation of the theory of moral sentiments left an ambiguous legacy, *prima facie*, ill at ease with the geopoetic impulse, as seen in the work of MacDiarmid and White and their illustrious predecessors.[45] We may, however, point to an immediate and important point of connection.

We have argued that *On a Raised Beach* can be understood as an epic of transformation that is grounded in MacDiarmid's embodied access to a liminal state, an encounter that he experienced as being outside of the constraints of the time order. Regarding temporality, this evocative poem can be understood as the analogue of the 'slow' or 'deep time' that has been valorised in contemporary ecological practice. By contrast, the apotheosis of political economy can be seen in the contemporary transformation of the global economic system through technological innovation. This transformation binds cost reduction and economic maximisation to the diminution of time and the computation of nanoseconds (Roberts 2001).[46] The intensified contradiction between an ecologically inspired deceleration and an acceleration that is ultimately traceable back to Adam Smith's endorsement of the division of labour greatly sharpens a longstanding conflict.

The progressive divorce of the economy from human participation serves as another 'posthuman' accelerator (Haraway 1991; Pepperell 2009). In sum, informational 'flow' became the 'flood' of 'big data'; the constant development and wider application of artificial intelligence (AI) renders ordinary human effort redundant; the individual human being becomes obsolescent and superfluous. We may construe the assertion of 'slow time' as an earth-grounded mode of resistance to be set over against 'acceleration' in the economy. In sharing in the simple moment of juxtaposition of when MacDiarmid's fingers held the stone, we now challenge the virtual cyborg and the accelerating spatio-temporal matrix.[47] In such acts, we reclaim our humanity and resist the sterile 'beyond' of the post-human condition. Others of course argue for the emancipated delights of technological innovation (Braidotti 2013). In a way that is distantly reminiscent of Luther's conception of the Christ Child in the stable crib bearing the weight of Divinity, we are, in effect, reduced to silence. The theological implications are considerable, and we can but hint at trails that might be followed up.

In principle, Roman Catholic theology would initially appear to be better equipped than Reformed traditions to develop a 'fundamental theology' that is able to admit and respond to the changing interface between evolving 'Tradition' (as for example brilliantly articulated by Yves Congar 1966)[48] and the transformed categories of space, time, and embodiment in a virtualised and accelerating world. In Reformed theology, there is a tension between Karl Barth's handling of the categories in relation to the theology of the all-comprehending Word[49] and the line of argument that is related to 'common grace' that

is pursued by Abraham Kuyper (1837–1905), Herman Bavinck (1854–1921), and Hendrikus Berkhof (1914–1995) in the Netherlands (Roberts 1975).

My own approach to the possible futures of theology is informed by, for example, William Gibson's cyberpunk classic *Neuromancer,* which was published in 1984 (Gibson 1984). Remarkably, it has been suggested by Jack Womack that *Neuromancer* influenced the development of the World Wide Web that was invented by Tim Berners-Lee in 1989. There is also evidence to suggest that cultural theory emerged from fiction. Moreover, I would suggest that underlying the virtuosic and inventive explosion of jargon and compressed narrative of *Neuromancer*, there lies the fundamental question as to whether cyborg enhancement advances or diminishes what it is to be human and humane in a post-human world (or worlds). In short and construed thus, William Gibson draws our attention back to the kenotic thread of theological reflection.

**Conclusion: Human Ecology and a New 'Education of Humanity'?**

Hugh MacDiarmid's austere epic *On a Raised Beach* confronts humanity with the intransigence of the rocks, yet it also implies a union between the microcosm of the grasped pebble and the macrocosm of the Earth, indeed of the Cosmos as a globalised totality (Csordas 2009). In this paper we have maintained that this apparently simple juxtaposition of thumb, stone and finger was grounded in an experience which has affinities with contemporary moves in 'deep ecology'. Such commonality in experience allows for the exploration of a number of important issues surrounding the new dialectics of human and all life under the conditions of a global pandemic. How, then, might all these threads be drawn together?

Regarding global ecology, the mid- to late twentieth century can be regarded as deceptively innocent years, when the heroic efforts of Rachel Carson in *Silent Spring* (Carson 1964) and Carolyn Merchant's pioneering ecofeminism in *The Death of Nature* (Merchant 1980) were early warnings and accounts of the impending environmental crisis that was understood as being caused by human intervention were attacked as propaganda (Oreskes and Conway 2012). There is now a global climate emergency that is recognised by a significant majority. Paradoxically, however, the new-born desire for mass immersion in nature rather than the earlier ventures into the residual natural world by elite individuals or self-selected minorities gives rise to its own problems. For a well-publicised example, the highest mountain in Wales (Snowdon, or Yr Wyddfa in Welsh) is now subject to heavy erosion, with some 700,000 visitors each year.[50] The need for an all-embracing 'human ecology' has moved from the periphery to the core of urgent concern. The work of earlier figures such as the polymathic pioneer of town planning Patrick Geddes (1854–1932), and, for example, the writings and activism of such figures Alastair McIntosh (McIntosh 2001) and Michael Northcott (Northcott 2013) have been exemplary in Scotland.

In this paper we have argued that Hugh MacDiarmid's poem *On a Raised Beach* may be appropriately understood as the early rebirth on Scottish ground of transformative direct engagement with the Earth. In this great poem, which was born of an altered state of consciousness (ASC), there swirl geology, shamanic journeying, a critique of derogate patriarchal theology, traces of Christ in the wilderness, the dialectics of *instasis* tending to non-duality, the implicit touch of Gaia—and the project of the renewal of Scotland. MacDiarmid's epic may help us to grasp and begin to understand that the task of *geopoesis*, world-making, which is intrinsic to Scottish sensibility.

Stripped of illusions and pretensions, we too may seek the sacred in the particular that touches the whole. The search for this specificity is open to anyone who grasps a pebble *with full awareness*. John Buchan and R. D. Laing, as well as MacDiarmid, sensed the wrath of Kali/Gaia in different contexts and in mythic terms. We will have to pass through the terror before we are able to touch tenderness once more. For humanity, this will mean passing through pain and anger into grief—and possible renewal.

It would appear that virtual encounters with 'the real' are increasingly deemed insufficient by millions, but this search for identity through 'nature' may well destroy the object

of our devotion and rough love. Comprehensive symbolic renewal is required if what James Lovelock characterises as the 'human plague' (Lovelock 2019) is to turn away from what Kenneth White has called the 'mass-mess' and to find renewal in the 'new normal' of ongoing pandemic caution.

This will be life lived in reduced circumstances. What this reading of Hugh MacDiarmid's great poem ventures to imply in the longer term is that there are viable ways of living more fully with much less once humankind re-roots itself in practices that have been long excluded from our individual and collective 'makings of humanity'. We might well begin this journey with an intensified and informed awareness of the full significance of the pebble held between the fingers and the thumb.

> Yet will creation turn to thee
>
> When, love being perfect, naught can die,
>
> And clod and plant and animal
>
> And star and sky,
>
> Thy form immortal and complete,
>
> Matter and spirit one, acquire,
>
> *Ceaseless till then, O Sacred Shame,*
>
> *Our wills inspire!*
>
> From *Hymn to Sophia: The Wisdom of God*,
>
> Hugh MacDiarmid, *The Complete Poems*, Vol. 1, p. 455.

**Funding:** This research received no external funding.

**Conflicts of Interest:** The author declares no conflict of interest.

## Notes

[1]　Accessible at http://www.geopoetics.org.uk/geopoetics-in-a-time-of-catastrophic-crisis-the-fourth-tony-mcmanus-lecture-by-richard-roberts/ (accessed on 2 December 2021).

[2]　My close involvement in the Centre for Human Ecology in Edinburgh in the late 1990s exposed me to both the complexity of the issues involved and to the resonance and 'elective affinities' (a term derived from Goethe and theorised by Max Weber) between ecological and human trauma.

[3]　In this, White is far from unique: Lewis Grassic Gibbon (1901–1935), author of *A Scot Quair* (1932–1934), died as it were in exile in Welwyn Garden City; Charles Rennie Mackintosh (1868–1928), the renowned architect, designer, and artist suffered a similar fate. Both enjoy a vibrant ongoing posthumous reception.

[4]　An explanation unsourced by McManus but crucial to his presentation of Kenneth White's work.

[5]　Scotland remains a place in which local knowledge is highly valued. The extinction of place through development is less advanced than in England, not least as a consequence of lower population density (70 people per square kilometre, as opposed to 270 per square kilometre in England). MacDiarmid's experience informing *On a Raised Beach* is not undercut by urban development in the way that any reading of Richard Jefferey's (1848–1887) *Bevis: the Story of a Boy* (1882) is now fraught with nostalgia for lost landscapes.

[6]　Alastair McIntosh's highly successful *Soil and Soul: People versus Corporate Power*, (McIntosh 2001) is another example that resonates with the geopoetic genre. This text recounts the dramatic story of pioneering resistance to the threatened destruction through quarrying of an entire Hebridean mountain.

[7]　Those who are familiar with MacDiarmid's earlier masterpiece in 'synthetic' Scots, *A Drunk Man Looks at the Thistle* (1926), will be aware of the impact of Spengler on the poet.

[8]　The ground in popular Evangelical Christian culture in the United States was prepared through, e.g., the highly influential writings of Hal Lindsey (1970), *The Late Great Planet Earth*. It is noteworthy that in his recent brilliant Gifford Lectures (2020–2021), *Networks, Nodes, and Nuclei in the History of Christianity, c. 1500–2020*, Professor David Hempton explored the longstanding and problematic association between crises and Protestant pre-millennialism.

[9]　A term relished by MacDiarmid, originally theorised by the Scottish literary critic George Gregory Smith in his *Scottish Literature, Character and Influence* (Smith 1919) as a way of rendering explicit specific the distinctiveness of Scottish Literature and resisting its assimilation into English Literature.

10  Figures such as Lord Rowan Williams, the late Bishop Stephen Sykes, Professor David Ford, among others have acknowledged the formative impact of MacKinnon.

11  MacDiarmid's masterpiece in Scots, *A Drunk Man Looks at the Thistle* (1926), is indicative of this paradox: the revival of Lallans (Lowland Scots) was at odds with the ongoing necessity of writing in English; that is in, as it were, the language of the enemy. There is of course a long and continuing culture-political on the status of Scots as a language fully distinguishable from English or a conserved dialect of the latter.

12  *On a Raised Beach* is both an eruption and an irruption, i.e., external expression and internal revolution.

13  This competition is alluded to by Kenneth Buthlay in the Preface to *A Drunk Man Looks at the Thistle* and his highly informative, indeed indispensable critical commentary on the text (Buthlay [1987] 2008, vii, pp. 78–79).

14  During the author's time as Professor of Divinity at the University of St Andrews, his duties included the assessment of a postgraduate essay prize competition. An aspirant geologist traced the antagonism between Scotland and England back to continental plate migration, which had had the unfortunate result in the misplaced juxtaposition of an ancient Hercynian Scotland with an England that had a different, less noble primordial geological ancestry. Needless to say, plate tectonics is a complex geological topic, but the mythological point was well made.

15  In the course of a conversation with the late Tessa Ransford and the leading Scottish politician Michael Russell, held prior to the 2014 Scottish Referendum, the latter referred to Scottish Independence as an 'escatological hope'.

16  All citations of Hugh MacDiarmid's poems are to (MacDiarmid 1978).

17  There is extensive discussion of the communicative purposes of *Verfremdungseffekt*. The Russian origins are explored in Robinson (2008).

18  Jean-Paul Sartre's 1940 essay written under the influence of Husserl, *The Imaginary: A Phenomenological Psychology of the Imagination,* initiated prolonged usage and development of this concept.

19  This observation opens up another channel of enquiry in interpreting nature mystical experiences in which such terms as 'non-dualism', neo-vitalism, pantheism, panentheism, etc., are deployed.

20  MacDiarmid's historic and attitudinal distance from the present flood of women's writing and poetry in Scotland is expressed in his observation that: 'Scottish women of any historical interest are curiously rare . . . our leading Scotswomen have been . . . almost entirely destitute of exceptional endowments of any sort.' See Roberts (2008).

21  See 'Poets' Pub', National Galleries of Scotland, Available online: https://www.nationalgalleries.org/art-and-artists/8217 (accessed on 9 July 2021).

22  The original French text is at https://www.amisdeproust.fr/images/DocsPdf/la_madeleine.pdf, (accessed on 29 October 2021).

23  Available online: http://art.arts.usf.edu/content/articlefiles/2330-Excerpt%20from%20Remembrance%20of%20Things%20Past%20by%20Marcel%20Proust.pdf, (accessed on 29 October 2021).

24  The use of the term 'shaman' (*šamán*) and its cognates raises many questions. Whilst the etymology of the word 'shaman' was traced in anthropological studies of Siberian shamanism, the classic comparative study by Eliade, Mircea (Eliade 1964), *Shamanism: Archaic Techniques of Ecstasy*, Princeton, NJ: Princeton University Press is contested, and contributors like the anthropologist turned practitioner Michael Harner (Harner 1980), Ioan M. Lewis (1971), Kenneth Meadow (Meadow 1991), and the distinguished anthropologist Geoffrey Samuel (Samuel 1983) besides and many others, serve to show that whilst 'shamanism' is not as such an essentially contested concept, it is nonetheless capable of wide application. The ascription of 'shaman' to Kenneth White's role of the grounded 'worlding' writer tasked with a 'journey' takes up some elements present in this discussion. The appropriation of the term is resisted by indigenous peoples' activists seeking to preserve their distinctive cultures, see the *Psuedo Shamans Cherokee Statement* at http://www.thepeoplespaths.net/Articles2001/RLAllen-CherokeeStatement-Shamans.htm, (accessed on 30 November 2021).

25  Kenneth White's articulation of shamanism appears in poetic form in, to take but a single example, 'The Shaman's Way' (White 2003, pp. 187–89).

26  It is difficult now to convey without anecdotage the drama that would on occasion attend university teaching over a half century ago, before the 'diamonds into glass' normalisation that took place following the Education Reform Act of 1988. This made postgraduate life exciting: one never quite knew who might be smitten. Once gained, unlearning such skills can be difficult.

27  I expand upon this in the course of a discussion of the social construction of the concept of 'nature' in Roberts (2004).

28  John Muir (1838–1914), the Scottish-born pioneer ecologist and campaigner for national parks in the United States is now celebrated in his birthplace Dunbar. Whereas Yosemite became a national park in 1890, Scotland's parks, Loch Lomond and The Trossachs National Park (created in 2002), and the Cairngorms National Park (created in 2003) were designated over a century later. This failure of connection is significant. In Irvine Welsh's novel Trainspotting (1996) a key scene is when the characters Renton, Spud, Sickboy, and Tommy encounter Scottish wilderness for what appears to be the first time at the remote Corrour Station, on the northern edge of Rannoch Moor. They are much affected. This location is revisited in the *T2 Trainspotting* (2017). This is alienation from nature writ large. Organisations like the GalGael Trust in Govan seek to reverse this alienation through training and community development.

29  This is apparent when repeated political demands are made to invest ever more money in responding to Scotland's drug and alcohol abuse problem, proportionately the worst in Europe, rather than to seek out in anthropological terms the socio-cultural dynamics that fuel this dire reality.

30  Key issues in the debate are summed up by Liam Connell (Connell 2003), in a contribution informed by (amongst others) Edward Said in *Culture and Imperialism* (Said 1993), which owes in turn an explicit debt to Hegel. I explored this connection in Roberts (1988). Regarding Scotland and Anglocentric hegemony, this is a complex and contentious area that invites repeated investigation, not least given changes in Scotland with devolution and the restoration of the Scottish Parliament after the 1997 referendum.

31  Alan Bold's account of the tense interactions of between Willa Muir and Valda Greave with MacDiarmid affords a further glimpse into the question of embodiment. See Bold (1990, p. 381). Willa Muir's *Women: An Inquiry* (Hogarth Press, 1925) and *Mrs Grundy in Scotland* ("The Voice of Scotland" series), Routledge, 1936, are indicative of her advanced thinking. Wider questions lurk here regarding the emergence of the post-Reformation 'body' in Scotland. See Peter Brown (1988), *The Body and Society: Men, Women and Sexual Renunciation in Early Christianity* (Columbia, 1988), and, from the Jewish side, Daniel Boyarin, *Border Lines: The Partition of Judaeo-Christianity* (Divinations: Rereading Late Ancient Religion) (University of Pennsylvania Press, 2004) on the tormented origins of Christian attitudes to sexuality as rediscovered and re-intensified in the Scottish Reformation and its long aftermath.

32  This was written just as the 2021 COP26 conference in Glasgow was drawing to its conclusion.

33  My extensive participant observation fieldwork and teaching theories of shamanism and ritual in Religious Studies at Lancaster University and at Stirling lie behind these proposals.

34  The brilliant and controversial psychiatrist R. D. Laing is another figure of importance here. See Clay (1997).

35  At this juncture, contested questions once more arise about the 'cultural appropriation' of concepts and practices claimed by indigenous peoples. My point is simply to point to parallels and affinities, and beyond this, to point out that the global crisis requires humanity as a complex totality to recognise that it must re-learn lost indigeneity, or, as James Lovelock prophesises, face extinction.

36  The consequences of the longstanding absence of competently managed and effective rites of passage and calendric ritual in modernity are addressed in, for example, Roy Rappaport's, *Ritual, Religion and the Making of Humanity* (Rappaport 1999). Is humankind equipped to evolve without such practices?

37  This systemic immaturity and the emergence of what the late Robert Bly called the 'sibling society' in which adolescents fail to become adults (Bly 1997) is recognised as the 'kidult' in modern marketing. See Jim Carroll (Chairman of BBH London), 'The man with the child in his eyes: has modern marketing become infantilised?'. Carroll argued 'that in the pursuit or creativity and collaboration, marketing has become too childlike and forgotten the serious nature of the business', at https://www.campaignlive.co.uk/article/man-child-eyes-modern-marketing-become-infantilised/1312191, (accessed on 2 December 2021).

38  See the Third Lecture, 'Gaia a finally secular figure for nature'.

39  I first used this term thirty years ago in the agenda for the conference *Religion and the Resurgence of Capitalism* held at Lancaster University in 1991. The final theme was: 'An Ecological Eschaton?—Religion between Capitalism and Constraint—What kinds of response are world religions likely to make to the inescapable tension between demands for economic growth and the coming "ecological crisis"?'. https://core.ac.uk/download/pdf/229433887.pdf, (accessed on 14 July 2021).

40  The prestigious Gifford Lectureships were established by Adam Lord Gifford (1820–1887), a senator of the College of Justice in Scotland. The purpose of Lord Gifford's bequest to the universities of Edinburgh, Glasgow, St. Andrews, and Aberdeen was to sponsor lectures to "promote and diffuse the study of Natural Theology in the widest sense of the term—in other words, the knowledge of God'. https://www.giffordlectures.org, (accessed on 13 July 2021).

41  https://www.winterwatch.net/2019/01/james-lovelock-and-the-anti-human-gaia-subterfuge/. (accessed 19 October 2021).

42  Kenneth White's poem 'The Shaman's Way' is indicative of his approach to shamanism (White 2003, pp. 187–91).

43  The observations made here are confined to the western setting for reasons of space. The creative potential of VR for Buddhism is celebrated by Deepak Chopra, as reported by Sam Dean (Chopra/Dean undated).

44  At this point in the lecture delivered virtually on Zoom, participants were invited to hold a pebble and reproduce for themselves the sensory correlate of the opening passage of *On a Raised Beach*.

45  The lecture out of which this paper has been developed was originally destined to be delivered in Panmure House, Adam Smith's residence in Edinburgh, and so references to Smith were in order. A full confrontation between MacDiarmid and Smith, and more widely between literature and political economy, would have required at the very least another lecture. The names of William Blake (1757–1827), William Wordsworth (1770–1850), and John Ruskin (1819–1900) as well as that of D.H. Lawrence (1885–1930) become relevant in this connection.

46  The writer first visited this topic in Roberts (2001).

47  Interestingly, in a later work, Michael Heim welcomes and celebrates virtual and cyborg enhancement (Heim 2000).

48  Tradition, *Tradition and Traditions*, London: Burns & Oates; (Congar 1966).

49  My exploration of the category of time began with my doctoral research. The approach is summarised in Roberts (1992).

50  BBC News 23 August 2021, 'Walkers on Snowdon have been urged to respect the mountain amid concerns over the impact of a spike in visitors', at https://www.bbc.co.uk/news/uk-wales-58283816, (accessed on 29 November 2021).

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
