# Peer review of "Hugh MacDiarmid’s ‘On Raised Beach’: ‘Geopoetics’ in a Time of Catastrophic Crisis’â€"

_religions, doi:10.3390/rel13010031_

Round 1
Reviewer 1 Report
This is an exhilarating read with its array of references, ideas and themes. There are some minor typos (e.g. Irving Welsh) and perhaps some small adjustments needed as the essay transitions from conference paper to journal article. The conclusion might also be developed, perhaps by anticipation at an earlier stage of the argument. And I would be intrigued to learn more about the wider theological implications of this material. The essay also caused me to wonder what forms of political and religious institutional expression can be identified (esp. in Scotland) for the task of addressing the crises that are so eloquently identified here.
Author Response
Text previously revised with new Abstract submitted.
Reviewer 2 Report
This is a remarkable essay, unusual in style given its autobiographical character and yet it works as a scholarly essay of unusual interdisciplinarity and topical breadth. The poem of McDiarmid serves as a microcosm of overlapping local and global realities and to engage a remarkable range of other texts with erudition and insight. On possible changes given the mortality associated with unemployment and mental ill health (consequences of ‘lockdowns’) there is no empirical competition between economy and saving lives. So I think that claim needs adjusting in the light of the empirical data which clearly show internationally an association between stricter public health restrictions on normal life and raised mortality. There are two places where grammar needs correcting. One at lines152-3 and the other in the conclusion where there is an incomplete sentence that would work better asa clause joined to the previous sentence. Otherwise this paper is of great merit .
Author Response
The second review was replaced with the third which I am now dealing with (02/11/2021).
Reviewer 3 Report
This is an interesting and valuable contribution. The contribution is significant and the discussion erudite. As described below some organizational changes are probably warranted to clarify the structure, direction, and conclusions of the paper. Other minor changes in style and additional references will improve the paper as well.
Lines 4-23: Shorten abstract, and perhaps hone or narrow scope. The final two sentences of the abstract do not appear to be well represented in the content and conclusions of the paper.
Lines 24-26: Fewer keywords (10 max)
Line 47: Add a period after citation.
Lines 50-53: language is perhaps too vague.
Line 59: explain “experience of non-dualistic identification”, perhaps in FN
Line 63: correct word omission?
Line 64: perhaps cite brief reference(s) to Duns Scotus (and others?) for haecceitas.
Line 66-67: Why do these considerations draw us in this direction? Perhaps elaborate here and perhaps elsewhere. Or direct the reader to where this elaboration will occur.
Line 76: “been such as to require a mediator” is somewhat awkward.
Line 96: where do you cite it in full? Perhaps say where here?
Line 112: correct word omission?
Section 1: This section should perhaps more clearly articulate the direction of the paper. The section is quite long and does not direct the reader inescapably regarding what to expect in terms of sequence, logical flow, or conclusions. It might be worth splitting this section into multiple sections with a traditional (and rather short) introduction to the flow and conclusions of the paper, followed by one or more new sections (using existing section 1 material) which set the background for the subsequent work.
Section 2: The concluding portion of this section might work better as a conclusion to a shortened section 1, with parts of section 1 moved to current section 2 or the current material split into shorter, more clearly thematic sections.
Line 194: correct word omission?
Line 205: perhaps an alternate word for “multidimensional” would improve the sentence.
Lines 216-221: citation should be set as a block quote.
Lines 233-234: add reference(s) to key theoretical works?
Lines 238-241: sentence style can be improved.
Line 242: add reference(s) to works that epitomize this contestation?
Line 292: delete extraneous period.
Lines 338-344: if a block quote indent from surrounding text.
Line 386: replace “of” with “or”.
Lines 396-399: correct style.
Line 405: perhaps change word order to “may now in turn”.
Line 462-465: perhaps instead of serial “ands” in this sentence the structure could be altered to use a comma delimited list.
Line 480: perhaps instead of “will expose” the phrase “have the potential to expose” may be more defensible.
Line 492: replace “has” with “have”.
Line 513-514: correct sentence fragment.
Line 532: correct spelling.
Section 5: Greater explanation of shamanic hermeneutics in the paper, perhaps very early on, but certainly also in section 5 would be welcome.
Line 564: perhaps place “VR” in parentheses.
Line 575: replace “have” with “has”.
Line 591: I’m a little nervous about the regular descriptions of the present as obviously and egregiously bad (e.g. line 591: “our tormented present”) in light of the fact that most historians would suggest that, short of pre-agricultural existence, most people live better lives in most ways than they did in previous eras. There is no doubt that we are at risk of climate catastrophe, and that there is significant evidence of its beginnings , but to describe the current state of affairs as currently or obviously bad hamstrings the authors argument and will not convince those who perhaps most need convincing. The problem with our current state of affairs is that many of the problems are not obvious to the lay-person. A regular failing of environmentalists and climate activists is the use of proleptic language that treats a projected horrible future as if it is already obviously manifest. For most people it is not. Over-hyped language often results in dismissal of otherwise important arguments, which these are.
Section 7: As you mention at the outset, this section seems a bit out of place. The introduction should be deleted (or moved to a FN) and some editing of the section in its current form to adjust it more clearly to the overall thrust of the paper might be warranted.
Line 624: What are the considerable theological implications? The reader will be aided if these are laid out clearly.
Lines 628-632: Alteration of this sentence for smoother style may be warranted.
Lines 639-642: indent block quote.
Line 650: delete “may”.
Line 654: Perhaps a more delicate phrasing is warranted if the goal is to persuade.
Lines 657-658: it is perhaps more accurate to say that it “should move” to the core rather than that it “has moved” to the core. If it had moved to the core in general there would be more action.
Lines 671-673: correct sentence grammar.
Line 677: perhaps replace “men” with “humanity”.
I look forward to seeing the completed article!
Author Response
Close and extensive attention has been given to all the points raised by the third reviewer.